# Inhibition–Disruption of *Candida glabrata* Biofilms: Symmetrical Selenoesters as Potential Anti-Biofilm Agents

**DOI:** 10.3390/microorganisms7120664

**Published:** 2019-12-09

**Authors:** María L. De la Cruz-Claure, Ariel A. Cèspedes-Llave, María T. Ulloa, Miguel Benito-Lama, Enrique Domínguez-Álvarez, Agatha Bastida

**Affiliations:** 1Facultad de Odontología, Facultad de Ciencias de Bioquímica, Química Farmacéutica y Biología, Universidad Mayor, Real y Pontificia de San Francisco Xavier de Chuquisaca, Sucre, Bolivia Calle Dalence Casilla 497, Sucre 685041, Bolivia; 2Facultad de Ciencias de Bioquímica, Química Farmacéutica y Biología, Universidad Mayor, Real y Pontificia de San Francisco Xavier de Chuquisaca, Sucre, Bolivia Calle Dalence Casilla 497, Sucre 685041, Bolivia; cespedes.ariel@usfx.bo; 3Programa de Microbiología-Micología, Facultad de Medicina, Universidad de Chile (Av. Independencia 1027, Comuna Independencia), Santiago 7591538, Chile; mtulloa@med.uchile.cl; 4Consejo Superior de Investigaciones Científicas, Instituto de Química Orgánica General, (IQOG-CSIC) c/ Juan de la Cierva 3, 28006 Madrid, Spain; miguelbenitodelama@gmail.com (M.B.-L.); e.dominguez-alvarez@iqog.csic.es (E.D.-Á.)

**Keywords:** *Candida glabrata*, denture, candidiasis, biofilms, disruption biofilms, antifungal drugs

## Abstract

*Candida glabrata* is one of the most prevalent pathogenic *Candida* species in dental plaque on tooth surfaces. *Candida* biofilms exhibit an enhanced resistance against most antifungal agents. Thus, the development of alternative more potent and effective antimicrobials is required to overcome this resistance. In this study, three novel fluorinated derivatives and nine selenoester compounds were screened as novel antifungal and antibiofilm agents against *C. krusei*, *C. parapsilosis*, and *C. glabrata* (N = 81 dental isolates). *C. glabrata* strains were susceptible only to fluorinated compounds while *C. krusei*, *C. parapsilosis*, and *C. glabrata* were susceptible to the action of the selenoesters. The evaluated symmetrical selenoester compounds presented very good antifungal activity against all the tested *C. glabrata* dental isolates (1–4 μg/mL of minimum inhibitory concentration-MIC). The most active compound (Se-5) was able to inhibit and disperse *C. glabrata* biofilms. These results demonstrated that selenoesters may be novel and promising biocide agents against *C. glabrata* clinical dental isolates.

## 1. Introduction

The oral cavity can be colonized by different groups of microorganisms, such as bacteria, viruses, and fungi [1,2]. Bacterial adhesion to surfaces and the subsequent production of extracellular polymeric substances prompts the formation of surface-related bacterial communities called biofilms [3]. Biofilm infections can be caused by a single microbial species or by a mixture of species [4,5]. The most common infection by *Candida* is denture stomatitis, which is caused by the formation of biofilm on the surface of the acrylic denture [6]. The composition of the denture biofilm microbial community is similar to that of dental plaque, with the exception of an increase in *Candida* spp. [7]. Dental biofilms are composed of microorganisms embedded in an extracellular matrix that consists of organic constituents (proteins, lipids, carbohydrates, and glycoproteins) and inorganic constituents (calcium, phosphorus, sodium, potassium, and fluoride) derived from saliva [8,9] (Figure 1). The microorganisms growing in biofilms have a considerable impediment to antimicrobial therapy due to the specific architecture of the biofilm, nutrient limitation, slow growth, and overexpression of multidrug-resistance (MDR) efflux pumps [10,11,12].

*Candida glabrata* has been isolated in removable dental prostheses as one of the most prevalent pathogenic yeast species among the *Candida* genus because it presents a higher affinity for the acrylic surface of dentures [13,14]. This yeast is able to adhere to the mucous surfaces and to stick to the acrylic resins of dental prostheses, being this adherence a crucial step in the generation of biofilms [15,16]. The biofilm acts as a protective barrier that confers to its embedded fungal cells the ability to develop resistance towards the majority of the antifungal drugs commonly used in the treatment of chronic fungal infections [17,18,19].

The probability of developing candidiasis is higher in older patients because they present weaker defense mechanisms. Consequently, dentures predispose these patients to *Candida* infection, which was observed in 65% of patients in a previous study [20,21,22]. This pathogenesis is multifactorial because of its dependency of the host and of the yeast’s capability to adhere and proliferate in the host epithelial tissues. The antibiotics available to date are ineffective in the treatment of biofilm due to their high values of minimum inhibitory concentration (MIC) against *Candida* strains [23,24,25]. The reduced inhibitory or disruption effects against *Candida* biofilm exerted by current antifungal drugs and the rapid development of drug resistance have highlighted the need for the discovery of new antifungal agents [4,26]. A few selenium drugs have been described to present antimicrobial activity [27], but none of them has been described as a *Candida glabrata* biofilm inhibitor.

Regarding selenium, certain selenium derivatives or selenium nanoparticles have shown antifungal activity [28,29,30]. In this context, our group has reported in previous works the activity of selenoesters and selenoanhydrides against cancer cells [31], cancer multi-drug resistance [32], and resistant bacteria as *Staphylococcus aureus* and *Chlamydia trachomatis* [33], among other biological effects. The mechanisms of action of the selenocompounds are very diverse [34]. In the case of the selenium derivatives with anticancer properties, it has been reported, among other effects, that they can trigger apoptotic events [35], and have the ability to modulate the redox thiolstat thanks to the unique redox properties of this element [36]. Selenium can act, depending on the cellular environment, as an oxidant or a pro-oxidant. In contrast, not all microorganisms have enzymes to metabolize the selenocompounds. This lack of selenium metabolism enzymes can lead to the precipitation of nanoparticles of elemental selenium, which exert a toxic effect against pathogenic bacteria or fungus [37]. The formation of these nanoparticles is hindered in human cells because selenocompounds are metabolized within the cells, leading to the formation of H_2_Se. This key metabolite, depending on the cell requirements, can evolve to HSePO_3_^2-^, which is used to incorporate selenium in selenoproteins in the form of the selenocysteine aminoacid [38]. Alternatively, if there is an excess of selenium, it can be methylated to form dimethylselenide (excreted through lungs) or the cation trimethylselenonium, which is excreted through the urine [38]. This differential metabolism of selenium in human cells and in fungal/bacterial cells may enable the discovery of selective drugs towards these pathogenic microbial species.

The aim of this study was to describe novel fluoride and selenoesters drugs with antifungal and antibiofim activity against *Candida* strains commonly present in dentures. Thus, three fluoride derivatives and nine selenoester compounds were evaluated in vitro as inhibition/disruption biofilm agents against *Candida glabrata* isolates from dental prostheses of an adult care home in Bolivia. These selenoesters are structural derivatives of bioactive selenoesters reported in previous works.

## 2. Materials and Methods

### 2.1. Tested Compounds

The fluoride-containing compounds were supplied by Sigma-Aldrich España (Madrid, Spain) F-1 (o-(trifluoromethyl) phenethylamine) F-2 (m-(trifluoromethyl)phenethylamine) and F-3 (p-(trifluoromethyl) phenethylamine). Selenoesters were synthesized by Domínguez-Álvarez et al. [39]: Se-1 (*Se,Se*-bis(2-oxopropyl)benzene-1,4-bis(carboselenoate), Se-2 *Se,Se*-bis(methyloxycarbonyl methyl)benzene-1,4-bis(carboselenoate), Se-3 *Se,Se*-dibenzylbenzene-1,4-bis(carbo selenoate), Se-4 (*Se,Se*-bis(methyloxy carbonylmethyl)benzene-1,3-bis(carboselenoate), Se-5 (*Se,Se*-bis(2-oxopropyl) benzene-1,3-bis (carboselenoate), Se-6 (*Se,Se*-bis(cyanomethyl)benzene-1,4-bis(carboselenoate), Se-7 (*Se,Se,Se*-tris(cyanomethyl)benzene-1,3,5-tris (carboselenoate), Se-8 (*Se,Se,Se*-tris(cyanomethyl) benzene-1,3,5-tris(carboselenoate), and Se-9 (*Se,Se*-bis(cyanomethyl)benzene-1,3-bis(carboselenoate).

### 2.2. Growth Conditions

For the isolation of *C. glabrata*, Sabouraud Agar (DIFCO, NJ, USA) was used (glucose 40 g, agar 20 g, 0.05 g of chloramphenicol, and 1 L of H_2_O, at pH 6.5). CHROMID^®^ Candida bioMerieux microculture and API/ID32 Candida (bioMèrieux, France) and MALDI-TOF MS (bioMèrieux, France) were used. Susceptibility testing by disc diffusion agar used Mueller-Hinton agar supplemented with glucose 2% and methylene blue (5 µg/mL). Susceptibility of *C. glabrata* with the bioactive compounds was determined by the microdilution method in (Roswell Park Memorial Institute) RPMI medium -1640 (supplemented with MOPS, (3-(*N*-morpholino)propanesulfonic acid) at pH 7). FBS (fetal bovine serum 50%, bovine IgG <1 mg/mL hemoglobin <20 mg/dl), PBS (phosphate-buffered saline 0.01 M pH 7.4 Sigma-Aldrich, MO, USA), medium M 138 (peptone 2.5 g, glucose 5 g, yeast extract 2.5 g, malt extract 1.5 g, and chloramphenicol 0.05 g, 500 mL, and pH 6.5, adjusted with 0.1 N NaOH and 0.1 N HCl), Violet Gentian Solution (Sigma Aldrich, MO, USA), bleaching solution (25% acetone/75% propanol), and 24-well polystyrene plates (Biofilm, CA, USA) were used in the formation of biofilms. The microplate reader used was SpectraMax Plus 384 Microplate Reader. The optical density (OD) of the biofilms formed at 36 °C was measured after 24 h at OD 620 nm.

### 2.3. Isolation and Identification of Candida Strains

Three selected strains among the most pathogenic species of *Candida* genus were used as a control from the American Type Culture Collection (ATCC): *C. glabrata* ATCC 2001, *C. krusei* ATCC 6258, and *C. parapsilosis* ATCC 22019.

In total, *Candida* isolates from 81 adults from care homes (“Mercedes”, “Santa Rita” and “25 de Mayo”) from Sucre, Bolivia, each patient with informed consent] were employed in this study. The *Candida* strains were isolated by swabbing the surface of dentures of the upper or lower arch. [40 For the isolation of *C. glabrata*, agar dextrose Sabouraud and in CROMagar was used, in which *C. glabrata* had a mauve-pink color. The identification was carried out with API/ID32 *Candida* and MALDI-TOF MS.

### 2.4. Antifungal Assay

The antifungal activity of the tested compounds was evaluated by the agar disc diffusion (CLSI) and minimal inhibitory concentrations (MICs) methods [31]. The microorganisms kept at −80 °C were seeded onto Petri dishes with Sabouraoud dextrose agar culture medium supplemented with chloramphenicol (50 µg/L) and incubated at 37 °C for 48 h. The standardization of the inoculum for the susceptibility tests was carried out in 0.5 of McFarland, from five colonies of each microorganism (five reference strains of *Candida* species or the 37 isolated of dentures), which were streaked on Muller-Hinton agar supplemented with glucose 2% and methylene blue (5 µg/mL). Then, filter paper discs (6–10 mm), containing the tested compounds at appropriate concentrations (fluorinated compounds: 50–500 µg/mL; selenium compounds: 1–4 µg/mL), were placed on the agar surface. The inoculated plates were incubated for 24 h at 37 °C overnight for *C. glabrata* and *C. krusei* or 48 h for *C. parapsilosis*. The tested antimicrobial agents diffused into the agar and their inhibitions of *Candida* spp. growth were measured (Appendix A, Table 1). Sensidisks of voriconazole, caspofungin, and fluconazole were used as a control. The standardization of the inoculum for the susceptibility tests was carried out in 0.5 Mc Farland (0.5–2.5 × 10^6^ UFC/mL).

### 2.5. Minimal Inhibitory Concentrations (MICs)

The tested concentrations of selenoesters, voriconazole (VOR), fluconazole (FLZ), and caspofungin (CAS) were prepared in RPMI-1640 (pH = 7). The inoculum was prepared by suspending five distinct colonies (> 6–10 mm diameter) from 24-h cultures, in at least 3 mL of sterile distilled water. The inoculum was then suspended by vigorous shaking on a vortex mixer for 15 s, the cell density was adjusted to the density of a 0.5 McFarland standard, with the addition of sterile distilled water if required, to render a yeast suspension of 1 to 5 × 10^6^ colony forming units (CFUs)/mL. A working suspension was prepared by the dilution of the standardized suspension in sterile distilled water to yield 1 to 5 × 10^5^ CFU/mL. The 24-wellplate was prepared with 200 µL of cell suspension and 20 µL of selenoesters (1–100 µg/mL) or antifungal agents (0.2–10 µg/L for VOR, CAS, and FLZ) and incubated at 37 °C during 24 to 48 h. MIC was determined as the lowest concentration of the drug that inhibits the growth of the yeasts. The results were visualized by spectrometry at 530 nm.

### 2.6. Biofilm Inhibition Assay

The biofilm-forming ability of the *Candida* strains was measured using the crystal violet assay (CV), which is based on the ability of this dye to color the polysaccharide matrix [40,41]. In order to determine the ability of compounds to inhibit biofilms, the tested cultures were left growing in 24-well polystyrene microtiter plates in the presence of the respective fluoride or selenoester compounds at different concentrations. Briefly, each *Candida* spp. isolated from adult care homes (37 samples) and *C. glabrata* ATCC 2001, *C. krusei* ATCC 6258, and *C. parapsilosis* ATCC 22019 wild type were grown in 10 mL of rich M130 medium at 36 °C overnight in anorbital shakers at 150 rpm. Those cultures were diluted with M130 medium in order to obtain a final density of 1.0 × 10^6^ CFU/mL. Then, 0.2 mL of each strain were grown in a 24-well microtiter plate and incubated for 2 h at 36 °C until OD 620 nm was equal to 0.5 and added to the evaluated compounds (3 fluoride derivatives and 9 selenoester compounds) at different concentrations and sealed and left overnight at 36 °C. Three assays were performed for each strain in each plate. We included three controls: Negative control (vehicle, medium M138), without growth and without biofilm formation; positive control (vehicle, medium M138; and biofilm-forming strain); and positive control with antifungal drug (vehicle, medium M138; biofilm-forming strain and caspofungin). After biofilm surface priming, the medium in each well was removed carefully with a multichannel pipette, taking care to not disrupt the biofilm; each well was subsequently washed three times with PBS. The formed biofilm at the bottom of plate was stained with 0.5% aqueous solution of crystal violet for 3 min. After staining, plates were washed three times with sterile water and dried at room temperature. The optical density (OD) of each well was measured at 620 nm with an automated microplate reader SpectraMax Plus 384.

### 2.7. Biofilm Disruption Assay

In order to determine the ability of the fluoride or selenoester-containing compounds to disrupt pre-formed biofilms at the bottom of plate, the test cultures were formed in microtiter plate wells for 24 h. The pre-formed biofilms were then treated with the compounds at different concentrations for 24 h and the residual biofilm was estimated using the crystal violet assay. Microtiter plate wells containing fungal cells without antimicrobial compounds were used as controls during the experimentation. The data related to these experiments are depicted as the average values of the triplicate observations and error bars indicate the standard deviation (Appendix A). The average OD from the control wells was subtracted from the OD of all test wells. The biofilm inhibition assay and biofilm disruption assay were performed three times for all *Candida* isolate from the adult care home.

BBU (Biofilm Biomass Unit) = ODb 620 nm – ODc 620/ODb 620 nm

BBU (%) = Biofilm Biomass Unit by 100

ODb = Absorbance without tested compound

ODc = Absorbance with tested compound

BBU (t = 0) Biofilm Inhibition Units by 100

BBU (t = 24) Biofilm Disruption Unit by 100 

### 2.8. Data Analysis

The normality test was performed using the Shapiro Wilk test with a *p* value >0.05 with a 95% confidence level. Then, the results obtained were compared with the Tukey test using an ANOVA; the variance analyses were contrasted with the treatments of the fluoride and selenoester compounds according to the concentration and the time of formation of biofilms of the clinical strains of *C. glabrata*. Statistical comparisons among the different strains were performed by using Sigma Stat statistical software (SPSS, IBM, Armonk, NY, USA).

## 3. Results and Discussion

Among the 81 dental samples from the adult care home (>80 year (25%), 60–80 years (55%), and <59 years (20%)), the evaluated *Candida* species were found in 80% (Scheme 1).

Of the samples with *Candida*, 37 samples (58%) were identified as *Candida glabrata* and 70% of them had candidiasis (Scheme 1). In addition, among those 27 isolates with candidiasis by *C. glabrata*, 11 were able to form biofilm (Scheme 1, Appendix A). According to our data, older adults presented a 4.4-fold higher risk of developing candidiasis if their dentures were colonized by these microorganisms (prevalence 64,8%, IC 95% 1.727616–11.218251, and *p* = 0.0015) and more likely to cause candidiasis if colonized by *C. glabrata*. We observed an interesting behavior in mixed biofilms of *C. albicans* and *C. glabrata*, when compared to single biofilms. A 77% reduction of *C. albicans* was observed when this fungus formed mixed biofilms with *C. glabrata*. This reveals that there are competitive interactions in which *C. glabrata* predominates over *C. albicans* during the formation of biofilm. Those results verify that *Candida glabrata* is an emergent pathogen in the oral cavity present in dental stomatitis [42] and that it plays an important role in the formation of dental biofilms.

### 3.1. Antifungal Screening

Herein, we tested two types of compounds: (i) Fluorinated compounds and (ii) symmetrical selenoesters (Scheme 2). On one hand, the fluorinated compounds are small molecules bearing aromatic rings, trifluoromethyl, and amine groups. On the other hand, the symmetrical selenoesters contain a phenyl core substituted by two or three identical aliphatic moieties, each containing a selenoester and a functional group (methylketone, methyl oxygen ester, phenyl, or cyano) bound to selenium through a methylene linker. The chemical structure of the halogenated and selenated compounds is shown in Scheme 2. The fluoride compounds were available commercially, whereas the selenoesters were synthesized according to known literature methods [39].

The behavior of the fluoride compounds and the selenoesters against *Candida* spp. according to the agar diffusion method is reported in Table 1 (Appendix A).

To evaluate their antifungal activity, these molecules were challenged against drug-sensitive yeast *C. glabrata* and drug-resistant yeasts, such as *C. krusei* or *C. parapsilosis.* Results showed that fluorinated compounds were active against *C. glabrata* while Se-5 (ketone selenodiester), Se-7 (ketone selenotriester), and Se-8 (cyano selenotriester) had antifungal activity against all *Candida* species. Caspofungin (CAS), fluconazole (FLZ), and voriconazole (VOR) were used as a control.

The antifungal efficacy is expressed as the minimum inhibitory concentration (MIC), that is, the minimum concentration of the molecules required to inhibit the growth of the yeast (Table 2 and Appendix A). MIC values of selenoesters were determined against five reference strains and 37 isolate dentures. The obtained MIC values were compared against antifungal agents (voriconazole, caspofungin, micafungin, amphotericin B, and fluconazole) (Table 2 and Appendix A). MIC values of 37 dental isolates with the selenoesters ranged from 1 to 64 μg/mL against *Candida* spp.

In general, all selenoester compounds showed a broad-spectrum activity against various drug-resistant fungi (Table 2). The range of MIC values for the selenocompounds Se-5, Se-7, and Se-8 (-1,3-COSeCH_2_COCH_3_, -1,3,5-COSeCH_2_COCH_3_, or -1,3,5-COSeCH_2_CN) was 0.5 to 5 µg/mL against all commercial *Candida* species. Besides, Se-1 (-1,4-COSeCH_2_COCH_3_) exerted antifungal activity against the selected *C. glabrata* reference strain, with an MIC value of 8 μg/mL. 

As mentioned before, 11 dental isolates were able to form biofilms. Table 3 shows the MIC values of Se-5, Se-7, and Se-8 against those isolates and *C. glabrata* ATCC 2001. Standard antifungal agents, voriconazole and caspofungin were used for comparison. 

Selenocompounds Se-5 and Se-7 showed very good antifungal activity, with an MIC of 1 to 4 μg/mL or 1 to 16 μg/mL, respectively, for the 11 dental isolates whereas **Se-8** showed a moderate antifungal activity with an MIC value of 16 to 32 μg/mL (except with isolates 3 and 18, where MIC values were 4 and 8 μg/mL, respectively).

The above results suggest that an optimum substitution in the aromatic ring of di- or tri-esters is required for maximum activity. Thus, the symmetrical selenodiesters or selenotriesters could be good candidates as antifungal agents against *Candida* present in dentures. Among them, methylketone-containing selenoesters showed the most promising activities.

### 3.2. Inhibition of Biofilm Growth and Disruption of Preformed Biofilm

*Candida* species present the ability to form biofilms, which protects them from the action of antifungal drugs. As is well known, cells in biofilms on an abiotic surface are 1000-fold more resistant to conventional antibiotics than in planktonic cultures [44]. The general protocol used for the inhibition or disruption of the biofilm by fluoride derivatives or selenoester compounds against *C. glabrata* is represented in Figure 2. Here, IC_50_ or IC_90_ are defined as the concentration of the compound that inhibits the biofilm development by 50% or 90% (BBU = 0.5 or 0.9) while EC_50_ is defined as the concentration of compound that disperses 50% of a preformed biofilm (Appendix A).

All fluoride compounds (F-1, F-2, and F-3) presented a similar behavior against the biofilm formation generated by *C. glabrata* ATCC 2001 or dental isolates (Figure 2). The IC_50_ values [45,46] of F-1, F-2, and F-3 against *C. glabrata* reference strains and dental isolate biofilm formation were found to be 50, 100, and 150 μg/mL (t = 0), respectively (Figure 2 and Figure 3, the mean values of BBU for *C. glabrata* ATCC2001 with those compounds are shown in Appendix A). The IC_50_ value of caspofungin against *C. glabrata* dental isolates was found to be 1 μg/mL (red line, Figure 2). In the presence of fluoride compounds, the biofilm formation by *C. glabrata* decreased by 83% (*p* < 0.05), with most of the dental isolates at a concentration of 160 μg/mL (t = 0). This is consistent with the value described in the literature for fluconazole at the same concentration [47]. The IC_90_ value of caspofungin (t = 0) was 2 μg/mL and F-1, F-2, and F-3 were found to be 150, 300, and 320 μg/mL, respectively (Figure 2 and Figure 3). IC_90_ values at t = 24 h for all fluoride compounds were >320 μg/mL.

The discovery of novel antifungal agents with the ability to not only inhibit biofilm formation but also to disperse established biofilms would be a significant advance to tackle infections caused by *Candida* spp [48]. In order to evaluate the efficiency of fluoride compounds to eradicate preformed biofilms, F-1, F-2 and F-3 were used against *C. glabrata* ATCC2001 and against 11 dental isolates biofilms. Pre-formed biofilms of *C. glabrata* ATCC2001 were treated with six different concentrations of fluoride compounds, 50 to 320 μg/mL (Figure 3 and Figure 4; range 50–320 μg/mL). The EC_50_ values for F-1, F-2, F-3, and caspofungin (red line) were 50, 150, >320, and 2 μg/mL, respectively.

The same study of biofilm inhibition or disruption was carried out with the active Se-5, Se-7, and Se-8 selenoester compounds (Figure 5, Figure 6 and Figure 7). Se-5 was found to be an effective inhibitor of *C. glabrata* biofilm formation as shown in Figure 5.

The IC_50_ and IC_90_ values at t = 0 of Se-5 and Se-7 against *C. glabrata* ATCC2001 and the dental isolates biofilm formation were found to be similar at 3 μg/mL (Figure 6, Appendix A). The IC_50_ and IC_90_ values at t = 24 h of Se-5 and Se-7 against *C. glabrata* ATCC2001 and dental isolate biofilm formation were found to be 1 to 4 and >4 μg/mL respectively. Mean values of BBU for *C. glabrata* with those compounds are shown in Appendix A.

The selenoester Se-5 presents better biofilm inhibition or disruption than caspofungin or the other selenoesters (Figure 6, Figure 7). Se-5, in many cases, dispersed up to 85% of the biofilm at 2 μg/mL, when caspofungin and Se-7 dispersed 58% or 30%, respectively (Figure 7). The above results indicate thus the ability of symmetrical selenoesters to disperse biofilms. The effect of Se-5 on the formation of *C. glabrata* biofilms at t = 0 h and 4 μg/mL was reduced by 33.33% (0.59 ± 0.12–0.97 ± 0.02 BBU) biofilm formation compared to caspofungin at 2 ug/mL (0.93 BBU ± 0.03) (positive control), being statistically significant (*p* < 0.0001) in the two biofilms. After 24 h, Se-5 at 4 ug/mL reduced the biofilm formation (0.16 ± 0.25–0.96 ± 0.05BBU) by 66.66%, with a greater effect compared to the positive control of caspofungin (0.56 ± 0.17 BBU), with a *p* < 0.0001 in eight biofilms and *p* > 0.05 in four biofilms (Appendix A).

On the other hand, Se-7 (t = 0, at 2 μg/mL) reduced biofilm formation on *C. glabrata* with a greater effect in 91.66% (0.80 ± 0.12–1.10 ± 0.01 BBU) in comparison to caspofungin at 2 μg/mL (0.93 BBU ± 0.03) (positive control), being statistically significant (*p* < 0.0001). After 24 h at 4 ug/mL, the Se-7 compound did not reduce the formation of biofilms (0.05 ± 0.01–0.36 ± 0.02 BBU) in 100% of the biofilms compared to caspofungin (0.56 ± 0.17 BBU), with a *p*-value < 0.0001. The IC_50_ value of Se-8 against *C. glabrata* ATCC2001 and dental isolate biofilm formation was 10 μg/mL. The mean value of BBU for *C. glabrata* with Se-8 is shown in Appendix A. The IC_90_ value of Se-8 against *C. glabrata* dental isolates was 20 μg/mL. Se-8 presented poorer biofilm inhibition or disruption when compared to the other selenoesters evaluated in antibiofilm assay or to caspofungin (red line, Figure 8).

The effect of Se-8 on the formation of biofilms generated by *C. glabrata* dental isolates at t = 0 or 24 h and 4 μg/mL was very weak (0.15 ± 0.03–0.80 ± 0.08 BBU, and 0.07 ± 0.03–0.51 ± 0.14 BBU, respectively), compared to caspofungin (0.93 BBU ± 0.03, 0.56 ± 0.17 BBU, respectively), being statistically significant, *p* < 0.0001. Selenocompound had the lowest biofilm-reducing effect at 0 and 24 h among the four selenocompounds evaluated in this anti-biofilm assay.

If we compare caspofungin and selenoesters at the same concentration of 2 μg/mL, Se-5 and Se-7 present a similar biofilm inhibition but Se-5 presets better biofilm disruption against the tested *C. glabrata* dental isolates. At this stage, the underlying mechanism of action of these novel selenoesters against *C. glabrata* is unknown and more experiments are needed to determine the ultimate mechanism behind the antifungal activity described herein. Selenocompounds and among them, selenoesters, as briefly mentioned in the introduction [31,32,33,34,35,36,37,38,39], have been proven in previous works to be able to exert a wide range of interesting biological activities, both in the anticancer and antimicrobial fields; for example, the modulation of the redox cellular state, scavenging of free radicals, increase of the oxidative stress, induction of apoptosis, and inhibition of efflux pumps, among others [31,32,33,34,35,36,37,38,39]. With the caveat that future works would need to prove it, we could hypothesize that one of several of these mechanisms could explain the observed activities for the symmetrical selenoesters presented in this work.

The abovementioned biofilm inhibition exerted by the Se-5 compound makes it a promising antibiofilm agent. Thus, it could be a good starting point in the design of new antibiofilm agents against *Candida* spp. present in dental prosthesis. Besides, it will be necessary to carry out additional studies to elucidate the mechanism through which the active selenoester Se-5 exerts its promising antifungal and biofilm inhibition activity. The inhibition and disruption of the biofilm with all test cultures was statistically significant, with *p* < 0.05 in the case of the treated cells compared to the untreated controls.

## 4. Conclusions

In summary, our findings support that *Candida glabrata* isolated in dental isolates is one of the most prevalent pathogenic yeast species of the *Candida* spp. Novel inhibitor and disruptor biofilm compounds against *C. glabrata* based upon the symmetrical selenoesters scaffold were evaluated herein. Among them, Se-5 and Se-7 showed good antifungal activity against all species studied as well as against all dental isolates. Fluoride-containing compounds presented IC_50_ values of 50 to 150 μg/mL and EC_50_ > 320 μg/mL. The most active compound, Se-5, presented an IC_50_ value of 2 μg/mL and an EC_50_ value of 2 to 4 μg/mL. These active compounds not only inhibited the formation of biofilm but also were able to disperse the established biofilms. Consequently, symmetrical selenoester compounds could be a novel promising target of research in terms of the development of novel anti-biofilms drugs.

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
