# Peer review of "Inhibition–Disruption of Candida glabrata Biofilms: Symmetrical Selenoesters as Potential Anti-Biofilm Agents"

_microorganisms, 2019, doi:10.3390/microorganisms7120664_

Round 1
Reviewer 1 Report
The minimum inhibitory concentrations for the reported compounds are high.
There is no hypothesis explained for the mechanism of action of these drugs against candida species.
The figures could be better. Controls with just the vehicle are missing.
Author Response
The minimum inhibitory concentrations for the reported compounds are high.
We respectfully disagree with the reviewer about the MIC are high. The MICs of the active symmetric selenoesters, Se-5 and Se-7, were quite low, as they showed values between 1 and 4 μg/ml. These values are noteworthy and significant, especially when compared with the commercial antifungal drug Fluconazol (2-32 mg/ml). These MIC values make worthy the publication of these results as these compound may have a potential future use in clinical therapy or in the night disinfection of the dental prostheses.
There is no hypothesis explained for the mechanism of action of these drugs against candida species…….
We thank the reviewer for this suggestion. Unfortunately, a satisfactory explanation/elucidation of the mechanism of action would require the performance of novel assays, that will be time-consuming, this would delay the publication of the article. Besides, we consider that the amount of biological results (as well as their relevance in terms of activity) and of experimental work justify the publication.
Instead of explaining the mechanism, we have enlarged the introduction to include a brief panorama of the mechanisms of action described so far for previously-reported selenoesters of our group and for selenocompounds. Besides, we have included a brief discussion of the potential mechanisms just before the conclusion (with the caveat that they are hypotheses). We hope that this will be enough for the reviewer at this stage. Of course, we will study more-in-depth the mechanisms of action in future works with these compounds; as we agree with the reviewer that not only the activity is important, the mechanism is also interesting.
The figures could be better. Controls with just the vehicle are missing..
We thank the reviewer for helping us to improve the quality of the presentation of the manuscript. The resolution of the figures has been improved and we have enlarged them so that they can be easier to understand for the readers.
Regarding controls, we have included three controls, as explained in Material and Methods:
-Negative control (vehicle, medium M138), without growth and without biofilm formation.
-Positive control (vehicle, medium M138; and biofilm-forming strain).
-Positive control with antifungal drug (vehicle, medium M138; biofilm-forming strain and caspofungin).
All the typos, grammar and clarification issues listed above have now been amended. English language and style has been carefully revised by a researcher married with a native speaker, living thus in an English-speaking familiar environment.
We thank the reviewer for the feedback and for their constructive criticism, which we all feel that have greatly improved our contribution. The revised manuscript addresses all the queries and suggestions received from reviewer. All the changes to the text are highlighted using the tracked-changes function throughout the manuscript to ease their identification.
Reviewer 2 Report
The work of De La Cruz-Claire et al. it is well done from the experimental point of view, but some considerations must be included before publication.
The main concerns to the isolation of samples of the 81 adults’ care home. Thera are not bacteria in the isolates? This is an important point because probably he biofilm formation in its natural environment is a mixture of microorganisms and some paragraphs including these considerations must be included.
Also it is a little bit surprising is that no data about the correlation of age of the patients with the samples of Candida glabrata for biofilm formation. “Among the 81 dental sample from adults’ care home (>80 years (25%), 60-80 years (55%) and <59 years (20%)), the evaluated Candida species, were found in the 80%. Of the samples with Candida, 37 samples (58%) were identify as Candida glabrata and 70% of them had candidiasis (Scheme I). In addition, among those 27 isolates with candidiasis by C. glabrata, 11 were able to form biofilm (Scheme I, Table S1).” Why the other 16 strain were not able to form biofilm?. Even more, of the 37 Candiad glabrata 27 Candidiasis and 10 carrier of Candida ssp (no candidiasis). No considerations are mentioned about those 10 strains
Author Response
The work of De La Cruz-Claire et al. it is well done from the experimental point of view, but some considerations must be included before publication.
We thank the reviewer for the nice evaluation of our work.
The main concerns to the isolation of samples of the 81 adults’ care home. Thera are not bacteria in the isolates? This is an important point because probably he biofilm formation in its natural environment is a mixture of microorganisms and some paragraphs including these considerations must be included.
The reviewer is right and we thank her/his comment. In the samples of the oral cavity was detected a large number of microorganisms abounding 17% the Gram + bacterial (Streptococcos spp, Staphylococcus spp) and to a lesser extent 3% Gram- (Enterobacteriaceae).
Also it is a little bit surprising is that no data about the correlation of age of the patients with the samples of Candida glabrata for biofilm formation. “Among the 81 dental sample from adults’ care home (>80 years (25%), 60-80 years (55%) and <59 years (20%)), the evaluated Candida species, were found in the 80%. Of the samples with Candida, 37 samples (58%) were identify as Candida glabrata and 70% of them had candidiasis (Scheme I). In addition, among those 27 isolates with candidiasis by C. glabrata, 11 were able to form biofilm (Scheme I, Table S1).” Why the other 16 strain were not able to form biofilm?. Even more, of the 37 Candiad glabrata 27 Candidiasis and 10 carrier of Candida ssp (no candidiasis). No considerations are mentioned about those 10 strains
We thank the reviewer for this correct concern. Yes, we have observed an association between colonization of C. glabrata in gingival tissues and subprosthetic candidiasis, but in this work the aim is to study, from a medicinal chemistry point of view, the antifungal activity of the evaluated compounds.
According to our data, older adults presented a 4,4-fold higher risk of develop candidiasis if their dentures are colonized by these microorganisms (Prevalence 64,8%, IC 95% 1,727616-11,218251, and p = 0.0015) more likely to cause candidiasis if colonized by C. glabrata. We have observed an interesting behavior in mixed biofilms of C. albicans and C. glabrata, when compared to single biofilms: a 77% reduction of C. albicans has been observed when this fungus formed mixed biofilms with C. glabrata. This reveals that there are competitive interactions in which C. glabrata predominates over C. albicans during biofilm formation.
All the typos, grammar and clarification issues listed above have now been amended. English language and style has been carefully revised by a researcher married with a native speaker, living thus in an English-speaking familiar environment.
We thank the reviewers for the feedback and for their constructive criticism, which we all feel that have greatly improved our contribution. The revised manuscript addresses all the queries and suggestions received from reviewers. All the changes to the text are highlighted using the tracked-changes function throughout the manuscript to ease their identification.
Round 2
Reviewer 1 Report
Thank you for addressing the concerns and for making the language corrections. The images can be a lot better. I personally feel these images are not publication quality but acceptable if you will have to repeat the experiments for better figures which is not a viable option.
Author Response
Thank you for your comment. We have improved the quality of the Figures 1-3 and of the Schemes I and II, improving in the last the chemical drawing. We hope that now the figures meet the standards. Biofilm figures have left as they were because they cannot be enlarged without enlarging significantly the manuscript. They are paired in groups of two figures, only alternative would be unpair them and draw one by one. However, this modification may add at least 2-3 pages to the manuscript, enlarging it too much at the current reviewing stage (minor revisions). This would also distort a lot the proportion between figures and text. If these modifications are not enough we would kindly ask you to indicate which figures need to be fixed and which modifications are required. Thank you in advance.
Reviewer 2 Report
The ms has been improved and can be published
Author Response
Dear Reviewer,
Thanks
Sincerely
Agatha